REGISTERED REPORT PROTOCOL

# Effect of prebiotics on growth and health of dairy calves: A protocol for a systematic review and meta-analysis

**R. Branco Lopes[1]ᵒ, C. Bernal-Córdoba[1]‡, E. D. Fausak◉[2]‡, N. Silva-del-Río◉[1,3]ᵒ***

**1** Veterinary Medicine Teaching and Research Center, University of California Davis, Tulare, California, United States of America, **2** Carlson Health Sciences Library, University of California Davis, Davis, California, United States of America, **3** Department of Population Health and Reproduction, School of Veterinary Medicine, University of California Davis, Davis, California, United States of America

ᵒ These authors contributed equally to this work.
‡ These authors also contributed equally to this work.
* nsilvadelrio@vmtrc.ucdavis.edu

## Abstract

Prebiotic supplementation has been studied as a means to enhance growth and health in dairy calves; however, results seem to be inconsistent across studies. The first objective of the future review is to identify, summarize, appraise, and discuss the current literature on the prebiotic supplementation for dairy calves. The second objective is to evaluate the effect of prebiotic supplementation on growth and health of dairy calves. Eligible studies will be non-randomized and randomized controlled trials in English, Spanish, or Portuguese that examined the supplementation of prebiotics to dairy calves (up to 6 months of age) and reported growth or health outcomes. The main growth outcomes will be average daily gain, feed efficiency, and main health outcomes will be fecal score and diarrhea incidence. A search will be conducted in Biosis, CAB Abstracts, Medline, Scopus, and the Dissertations and Theses Database with the support of a UC Davis librarian. Two reviewers independently will screen the titles and abstracts of retrieved studies. The screening of full manuscripts will be performed by one reviewer. The data extraction will be executed based on pre-tested forms. Risk of bias will be assessed using the Cochrane Risk of Bias 2.0 tool. If feasible, a random-effects meta-analysis will be conducted. Heterogeneity will be evaluated with $I^2$ statistic. If possible, publication bias will be investigated by using funnel plots, Begg's adjusted rank correlation, and Egger's test. The certainty of the evidence will be determined using the GRADE (Grading of Recommendations, Assessment, Development and Evaluations) approach. The proposed review will contribute to the current knowledge on prebiotic supplementation for dairy calves; this information may guide management decision at the farm level and identify gaps of knowledge to be addressed in future research. The protocol is registered in Open Science Framework (https://osf.io/ar5g2/) and available in the Systematic Reviews for Animals and Food (http://www.syreaf.org/contact/).

**Data Availability Statement:** All relevant data from this study will be made available upon study completion.

**Funding:** This project was supported by California Department of Food and Agriculture - Antimicrobial Use Stewardship Program. No additional external funding was received for this study.

**Competing interests:** The authors have declared that no competing interests exist.

## Background

In dairy production systems, successful calf rearing is of upmost importance for the long-term profitability of the operation. Growth rate and health status of dairy calves have been associated with cow's future reproductive and productive performance [1–3]. However, raising healthy calves under our current management systems is challenging. In the US, the morbidity and mortality of preweaned calves reach 38% and 5%, respectively [4]. Supplementation of feed additives is a commonly adopted management strategy for dairy calves [4], that aims to improve growth rate and immunocompetence during this critical stage of life.

Prebiotics are feed additives that have been associated with an enhancement of growth performance and health status by modulating gut microbiota [5]. The International Scientific Association for Probiotics and Prebiotics defined prebiotic as "substrate that is selectively utilized by host microorganisms conferring a health benefit" [6]. Currently, the most known prebiotics are carbohydrate-based, specially oligosaccharides as fructo-oligosaccharides (FOS), galacto-oligosaccharides (GOS) and mannan-oligosaccharides (MOS) [7]. However, recently other substances as polysaccharides (e.g., xylans, pectins, inulins) and non-carbohydrate compounds (e.g., polyphenols, polyunsaturated fatty acids) have also been considered as prebiotics [8–10].

The efficacy of prebiotic supplementation in dairy calves seems inconsistent in the literature. Some studies have suggested that supplementing prebiotics (MOS and polysaccharides) to calves improves average daily gain and feed efficiency and decreases fecal shedding of pathogens [11–13]. Moreover, a meta-analysis indicated that MOS (commercial product Bio-Mos®) supplementation for dairy calves resulted in improved body weight gain [14]. However, recent evidence suggests that prebiotic supplementation (MOS and polysaccharides) has no effect on health or growth performance of calves [15–17]. At the present, it is lacking a comprehensive and updated systematic review or meta-analysis on the effect of prebiotics on dairy calves' growth rate and health. Results from the proposed study are critical to increase our understanding on the effects of prebiotics on calves' performance and health status. Our results may guide future management decision at the farm level and identify gaps of knowledge to be addressed in prospective research studies.

## Objectives

The first objective of the future review is to identify, summarize, appraise, and discuss the current literature on prebiotic supplementation for dairy calves. The second objective is to evaluate the effect of prebiotic supplementation on the growth and health of dairy calves.

## Research question

The research question that will be addressed in the future was developed based on the PICO (population, intervention, comparator, and outcomes) elements [8]. The question will be: does the prebiotic supplementation affect the growth or health of dairy calves younger than 6 months?

## Methods

The reporting of this review protocol is in accordance to the Preferred Reporting Items for Systematic Review and Meta-Analysis Protocols (PRISMA-P) statement [18] and followed the guidelines for systematic review in animal agriculture and veterinary medicine [19]. The protocol is registered in Open Science Framework (https://osf.io/ar5g2/) and available in the SYR-EAF (Systematic Reviews for Animals and Food; http://www.syreaf.org/contact/), which is an

online repository for publication of review protocols studying animals and food. The PRIS-MA-P checklist is included in S1 Table. Although not intended, any amendments to this protocol will be documented and justified in the final review manuscript.

## Eligibility criteria

As the research question, the eligibility criteria were defined based on the PICO elements.

### Study design, characteristics, and population

The systematic review will include primary research studies, including non-randomized and randomized controlled trials, which are available in English, Spanish, and Portuguese. Observational studies will be excluded. Eligible studies must have investigated pre-weaned or weaned dairy calves (up to 6 months of age), with no restrictions on calves' breed or sex. No restriction on publication date or publication status will be applied; however, manuscripts lacking primary data will be excluded.

### Intervention and comparator groups

Eligible studies must have evaluated prebiotic supplementation, with no restriction for prebiotic type, dose, or supplementation duration. Research studies assessing prebiotics as therapy to control or treat diseases will be excluded. The prebiotic supplementation must have been compared to no intervention or placebo. Studies evaluating positive control groups (e.g., antimicrobials) as the single comparator will be excluded.

### Outcome measures

Studies must include at least one main or secondary outcome. For growth, the main outcomes will include average daily gain and feed efficiency, while secondary outcomes will be body weight, body traits (e.g., heart girth, wither height, hip width, or body length), dry matter intake and rumen development indicators (e.g., volatile fatty acids, ruminal pH, papilla length, and papilla width). For health, the main outcomes will comprise fecal score and diarrhea incidence, whereas assessments of serum metabolites (e.g., glucose, beta-hydroxybutyrate), immunoglobulins, cytokines, and mortality will be considered secondary outcomes. The main and secondary outcomes are defined in S2 Table. The prioritization of the growth performance outcomes was based on their impact on lifetime milk production [20]. The health outcomes were prioritized based on their association with gut health [21] and their frequency of use in calf health research studies [22].

### Information sources

After consultation with an experienced health and veterinary science academic librarian (EDF), the following electronic databases were selected: Biosis (Web of Science, 1926 to present), CAB Abstracts (CAB Direct, 1973 to present), Medline (PubMed, 1966 to present), and Scopus (Scopus, 1996 to present). Grey literature will be searched to find unpublished data using Dissertations and Theses Database (ProQuest, 1861 to present). The bibliography of relevant studies will be hand-searched by the first author.

### Search strategy

The search strategy will be designed with the assistance of an academic librarian (EDF). For this, the first author will select key words from relevant literature. The librarian will perform a keyword and subject heading citation analysis by finding hand-selected references in PubMed

and CAB Direct. Keywords and subject headings will be collected and compared with those already utilized. Yale MeSH analyzer will be also utilized to compare common Medical Subject Headings across articles for PubMed interface. Boolean operators "AND" and "OR" will be used to connect the search terms. Based on strategies built in PubMed and CAB Direct, EDF will translated the search to the other databases: Biosis, Scopus, and Proquest Dissertations and Theses. The refinement of the search strategy will be performed by the EDF and RBL. A preliminary literature search was conducted on October 28th of 2020; the results are described in S3 Table.

## Data management

The retrieved studies from the search performed in each electronic database will be exported to the reference manager F1000 (Faculty of 1000 Limited, London, UK) and duplicates will be removed. The de-duplicated results will be exported to the Covidence systematic review management software (Veritas Health Innovation, Melbourne, AU) for title and abstract screening. The data from the studies that meet the eligibility criteria will be extracted and entered onto a Microsoft Excel spreadsheet.

## Selection process

All manuscripts identified after each database search will be screened by two independent reviewers (RBL and CBC). Both reviewers have received systematic review training and have domain-specific knowledge on animal science. None of the reviewers will be blinded to journal or author names. Text-mining tools will not be used for any screening level. Firstly, the manuscripts' titles and abstracts will be screened. For this first screening, the following questions will be used to identify potentially eligible studies for inclusion in the review:

1. Does the title or abstract describe a study with dairy calves?

2. Does the title or abstract describe a study with prebiotic supplementation?

3. Does the title or abstract describe a primary intervention study?

4. Does the title or abstract describe one or more of outcomes for growth (e.g., average daily gain, feed efficiency) or health (e.g., fecal score, diarrhea incidence)?

For each of the screening questions the available answers will be "no," "maybe," and "yes." Studies will be excluded if both reviewers responded "no" to one of the questions. In addition, only citations with "maybe" or "yes" answers will be screened in the following step. Any disagreements between reviewers will be resolved by discussion, if no consensus is reached a third reviewer will be consulted (NSDR).

A full manuscript screening will be conducted by RBL on the remaining studies. This screening included the 4 previous questions, plus:

5. Is the study a controlled trial with negative control group?

6. Is the study written in English, Spanish, or Portuguese?

7. Is the prebiotic a supplementation strategy (not treatment or metaphylactic approach)?

8. Is the study population (dairy calves) equal or less than 6 months old?

During the full manuscript screening the answer available will be "no" and "yes." Studies will be excluded if RBL answer "no" for one or more of the questions. The exclusion reason

will be recorded at this screening level. A pre-test will be conducted in 30 abstracts to evaluate the clarity of the screening questions for each screening level.

## Data collection process

Studies that meet the eligibility criteria will be extracted by RBL into a Microsoft Excel spreadsheet. Data extraction forms will be designed based on previous studies, and pre-tested using 5 studies. Additionally, to assure the accuracy and completeness of the data collection, a second reviewer (CBC) will audit all the data entered into the extraction forms [23]. Study-level data will consist of journal name, language, country, author's name and affiliation, year of publication, and funding information. Population characteristics include animals' breed, sex, age, housing (individual or group), production system (conventional or organic), assessment of passive transfer, and herd type (commercial or research). Intervention and comparator data will include the description of comparator, prebiotic's commercial name, scientific name, concentration, dose, form of administration (e.g., whole milk, milk replacer), and duration of supplementation. For continuous outcomes (e.g., average daily gain), the following information will be extracted: number of experimental units for each treatment group, least square or contrast means for each treatment group, mean differences from control, unit of results, lower and upper 95% confidence intervals (CI), standard error, standard deviation, *P*-value, and the time point of each measurement. For dichotomous outcomes (e.g., occurrence of diarrhea), the following information will be extracted: number and proportion of positive experimental units per treatment group, total number of experimental units per treatment group, unit of results, odds ratio, relative risk, lower and upper 95% CI, *P*-value, and the time point of each measurement.

## Risk of bias assessment

The risk of bias will be assessed at the outcome level by RBL, using the Cochrane Risk of Bias 2.0 tool (Sterne et al., 2019) [24], with adaptations in the signaling questions to fit the animal science. The following domains of bias will be assessed: a) randomization process, b) deviations from intended interventions, c) missing outcome data, d) measurement of the outcome, and e) selection of the reported result. In the randomization process domain, the allocation sequence is unlikely that a farm worker or a researcher would have any treatment preference for a specific calf; thus, the question "will be the allocation sequence concealed until participants will be enrolled and assigned to interventions?" will be dropped. In the deviations from intended interventions domain, the question "will be participants aware of their assigned intervention during the trial?" will be dropped, as dairy calves are the population of interest; we will assume that in all trials calves were blinded. A similar approach has been described in a previous systematic review assessing risk of bias in livestock studies [25]. For each domain, the risk of bias will be classified as "high," "some concerns," or "low".

## Data synthesis

If more than 3 studies evaluate similar treatments and outcomes, a meta-analysis will be conducted. Meta-analysis will be performed in R 4.0.3 (R Foundation for Statistical Computing, Vienna, Austria) using RStudio version 1.3.1093 (RStudio Inc., Boston, MA) with the "metafor" package [26]. A random-effects model will be adopted; we will assume that not all intervention effects are the same. The weight of each study will be determined by the inverse variance method [27]. Heterogeneity between studies will be assessed by calculation of the $I^2$ statistic [28], which varies from 0 to 100% and will be interpreted as following: 0% to 40%: might not be important, 30% to 60%: may represent moderate heterogeneity, 50% to 90%: may

represent substantial heterogeneity, and 75% to 100%: suggests considerable heterogeneity [29]. If detected heterogeneity is higher than 50%, the possible sources of heterogeneity will be investigated through subgroup and sensitivity analyses. A sub-group analysis will be performed categorizing the studies as pre- or post-weaning and according to prebiotic supplementation dosage, type, and duration. If at least 10 manuscripts report similar treatments and outcomes, publication bias will be investigated using funnel plots, Begg's adjusted rank correlation, and Egger's test (P <0.10). If publication bias is detected, the "trim-and-fill" method will be used to estimate the extent of the bias [30]. If less than 3 studies meet the eligibility criteria or if the heterogeneity of the intervention and outcomes of the eligible studies is high results will be presented descriptively.

## Confidence in cumulative evidence

The certainty of the evidence will be assessed by one reviewer (RBL) using the GRADE (Grading of Recommendations, Assessment, Development and Evaluations) approach [31]. GRADE appraises the certainty of the body of evidence gathered in a systematic review. The evidence will be evaluated based on risk of bias, imprecision, inconsistency, indirectness, and publication bias.

## Discussion

The presence of microorganisms in the intestine of calves is considered a co-evolution process [32]. The calves' intestinal microbiota plays diverse functions such as protection against pathogens, activation of immune systems and, contributes to their nutrition [33]. Thus, it has been hypothesized that the manipulation of the microbiota through prebiotic supplementation may enhance the growth and health of dairy calves. To test this hypothesis, the future systematic review and meta-analysis will summarize and assess the effect of prebiotics on the growth and health of dairy calves. If prebiotics are found to impact the growth or health of dairy calves, the results from the future study may be useful guidelines for dairy consultants and veterinarians. In addition, the findings will contribute to the current knowledge on prebiotic supplementation in dairy calves and may be informative to guide future research. The proposed review has several strengths; it will adhere to the guidelines for systematic reviews in animal agriculture and veterinary medicine [34,35], the search strategy will be designed with librarian support in order to identify the highest number of available studies. To minimize bias, the screening of titles and abstracts will de performed indenpently by two reviewers, and pre-test forms will be used for each screening level. Some of the anticipated limitations of the present study are source of heterogeneity across studies. These could include the prebiotics' viability, type, and dosage, trials season, study design of primary litterature and settings in which the trials were undertaken [22].

## Supporting information

**S1 Table. PRISMA-P (Preferred Reporting Items for Systematic review and Meta-Analysis Protocols) 2015 checklist: Recommended items to address in a systematic review protocol.** (DOCX)

**S2 Table. Definition of main and secondary outcomes.** (DOCX)

**S3 Table. Preliminary electronic search string used to retrieve studies examining supplementation of prebiotics for dairy calves on October 28th of 2020.** (DOCX)

## Author Contributions

**Conceptualization:** R. Branco Lopes, N. Silva-del-Río.

**Funding acquisition:** N. Silva-del-Río.

**Methodology:** R. Branco Lopes, E. D. Fausak.

**Project administration:** N. Silva-del-Río.

**Resources:** E. D. Fausak.

**Supervision:** N. Silva-del-Río.

**Validation:** C. Bernal-Córdoba.

**Visualization:** R. Branco Lopes.

**Writing – original draft:** R. Branco Lopes.

**Writing – review & editing:** C. Bernal-Córdoba, N. Silva-del-Río.

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
