## [Decision Letter · Decision Letter 0]

4 Jun 2021

Effect of prebiotics on growth and health of dairy calves: a protocol for a systematic review and meta-analysis

PONE-D-20-34384

Dear Dr. Silva del Rio,

We’re pleased to inform you that your manuscript has been judged scientifically suitable for publication (even though two of the reviewers misunderstood the submission's nature and did not fully support this) and will be formally accepted for publication once it meets all outstanding technical requirements.

Kind regards,

Spyridon N. Papageorgiou, DDS, Dr Med Dent

Academic Editor

PLOS ONE

Journal Requirements:

1. Thank you for stating in your Funding Statement:

'This project is partially supported by California Department of Food and Agriculture - Antimicrobial Use Stewardship Program.'

Please provide an amended statement that declares *all* the funding or sources of support (whether external or internal to your organization) received during this study, as detailed online in our guide for authors at http://journals.plos.org/plosone/s/submit-now

Please also include the statement “There was no additional external funding received for this study.” in your updated Funding Statement.

Please respond by return e-mail so that we can amend your financial disclosure and competing interests on your behalf.

Reviewers' comments:

Reviewer's Responses to Questions

**Comments to the Author**

1. Does the manuscript provide a valid rationale for the proposed study, with clearly identified and justified research questions?

Reviewer #1: Yes

Reviewer #2: Yes

Reviewer #3: No

2. Is the protocol technically sound and planned in a manner that will lead to a meaningful outcome and allow testing the stated hypotheses?

Reviewer #1: Yes

Reviewer #2: No

Reviewer #3: Partly

3. Is the methodology feasible and described in sufficient detail to allow the work to be replicable?

Reviewer #1: Yes

Reviewer #2: No

Reviewer #3: Yes

4. Have the authors described where all data underlying the findings will be made available when the study is complete?

Reviewer #1: Yes

Reviewer #2: No

Reviewer #3: No

5. Is the manuscript presented in an intelligible fashion and written in standard English?

Reviewer #1: Yes

Reviewer #2: No

Reviewer #3: Yes

6. Review Comments to the Author

You may also provide optional suggestions and comments to authors that they might find helpful in planning their study.

Reviewer #1: This manuscript is a protocol for a systematic review and meta-analysis for the effect of prebiotics on growth and health of dairy calves. In overall, it is well written. The description of data synthesis is complete. This meta-analysis will be able to summarize the published data and provide useful information to guide raising dairy calves.

Line 227, should “results from” be “results for”?

Reviewer #2: I acknowledge well that there are numerous sorts of research papers conducting animal study to identify effects of prebiotic feeding to calves since the results seem to be inconsistent primarily due to variations of study conditions. Indeed I have not yet found out any meta analysis report on this criteria so is affirmative to authors' study theme which is worth further evaluation. On the other hand, because the approach they will take does not seem particular, I am still unsure that they will be able to provide any of significant outcome or novel finding to cover their study rationale and to be warranted as the registered report. Therefore, I would like to encourage authors to conduct a meta analysis as they designed here without endorsement and the journal will deserve it at the timing of submission as a regular manuscript.

Reviewer #3: The aim of this manuscript was to identify, summarize, appraise, and discuss the current literature on prebiotic supplementation for dairy calves. However, it only describe the protocol to follow in a future meta-analysis.

Although the protocol could be considered correct, it follows the scientific principles for conducting systematic reviews and could be an element of scientific interest, if it is not accompanied by the results it is only an empty protocol.

The performance of a systematic review and meta-analysis of the effect of prebiotic supplementation on dairy calves health and growth performance is very relevant and of great interest. This protocol is valuable and would be part of Materials and Methods section of the future study.

7. PLOS authors have the option to publish the peer review history of their article (what does this mean?). If published, this will include your full peer review and any attached files.

Reviewer #1: No

Reviewer #2: No

Reviewer #3: No

---

## [Editor Report · Acceptance letter]

18 Jun 2021

PONE-D-20-34384 

Effect of prebiotics on growth and health of dairy calves: a protocol for a systematic review and meta-analysis 

Dear Dr. Silva-del-Río:

I'm pleased to inform you that your manuscript has been deemed suitable for publication in PLOS ONE. Congratulations! Your manuscript is now with our production department. 

Kind regards, 

on behalf of

Dr. Spyridon N. Papageorgiou 

Academic Editor

PLOS ONE